# Lymph Node Stromal Cells: Mapmakers of T Cell Immunity

**DOI:** 10.3390/ijms21207785

**Published:** 2020-10-21

**Authors:** Guillaume Harlé, Camille Kowalski, Laure Garnier, Stéphanie Hugues

**Affiliations:** Department of Pathology and Immunology, School of Medicine, University of Geneva, 1206 Geneva, Switzerland; guillaume.harle@unige.ch (G.H.); camille.kowalski@unige.ch (C.K.); laure.garnier@unige.ch (L.G.)

**Keywords:** lymph node stromal cells, T cells, autoimmunity, alloimmunity, viral infection, cancer

## Abstract

Stromal cells (SCs) are strategically positioned in both lymphoid and nonlymphoid organs to provide a scaffold and orchestrate immunity by modulating immune cell maturation, migration and activation. Recent characterizations of SCs have expanded our understanding of their heterogeneity and suggested a functional specialization of distinct SC subsets, further modulated by the microenvironment. Lymph node SCs (LNSCs) have been shown to be particularly important in maintaining immune homeostasis and T cell tolerance. Under inflammation situations, such as viral infections or tumor development, SCs undergo profound changes in their numbers and phenotype and play important roles in contributing to either the activation or the control of T cell immunity. In this review, we highlight the role of SCs located in LNs in shaping peripheral T cell responses in different immune contexts, such as autoimmunity, viral and cancer immunity.

## 1. Introduction

T cell responses are crucial for immune surveillance and to protect our body against pathogen infections or cancer progression. Professional antigen-presenting cells (APCs), such as dendritic cells (DCs), present pathogenic or cancer antigens to activate T cells, which differentiate into effector cells, leading to the elimination of pathogens or tumors. T cells also play a crucial role in setting of B cell responses by promoting germinal center (GC) formation and allowing the development of efficient humoral immunity. Antigenic peptides presented by major histocompatibility complex (MHC) molecules at the surface of the APCs are recognized by the T cell receptor (TCR) expressed by the T cell. In order to avoid the development of autoimmune diseases, one important regulation of T cell responses is the elimination, or the inactivation, of developing T cells that would react to endogenous self-peptides in the thymus. Although the thymic T cell selection aims to delete developing T cells expressing a self-reactive TCR, this process is not flawless and some autoreactive T cells may egress the thymus to reach the periphery. Therefore, peripheral mechanisms of tolerance are necessary to inhibit the activation of autoreactive T cells. The main pathways of inactivation of autoreactive T cells in the periphery are their suppression by regulatory T cells (Treg) and the induction of anergy. The role of DCs in Treg induction and anergy has been extensively described [1]. However, other mechanisms further contribute to the modulation of peripheral T cell activation and outcome. In particular, over the past decade, novel functions of stromal cells (SCs) localized in second lymphoid organs (SLOs), such as lymph nodes (LNs), and in nonlymphoid tissues have been described and suggest that SCs directly regulate T cell responses in multiple immune contexts. A better understanding of the pathways these cells use to regulate T cell responses may lead to the identification of new therapeutic targets and possibly improve the treatment of immune-related pathologies, such as autoimmunity, graft rejection, viral infections or cancer. In this review, we provide an up-to-date review of our knowledge of how SCs shape peripheral T cell responses.

## 2. Lymph Node Stromal Cells Regulate T Cell Migration, Localization and Homeostasis

Lymph node stromal cells (LNSCs) are nonhematopoietic cells (CD45^−^) that structure the architecture of the LN, therefore promoting a site-specific environment that favors cell–cell interactions. Four main subsets of LNSCs have been described based on their expression or not of podoplanin (gp38) and PECAM1 (CD31). LNSC subtypes include blood endothelial cells (BECs, CD31^+^gp38^−^), lymphatic endothelial cells (LECs, CD31^+^gp38^+^), fibroblastic reticular cells (FRCs, CD31^−^gp38^+^) and double negative cells (CD31^−^gp38^−^) [2]. LNSCs constitute a network that is important for the organization of hematopoietic cells inside the LNs.

Recently, multiple subsets of LECs and FRCs have been identified based on their functions and localization in LNs. Single-cell mRNA sequencing of FRCs, which potentially differentiate from mesenchymal stromal cells [3], identified up to nine distinct FRC subsets in mouse LNs [2]. Among these subsets, six were well characterized and exhibited specific functions in impacting immune cells (Figure 1). Marginal reticular cells (MRCs) are MadCAM1^+^, adjacent to subcapsular sinus, and produce CXCL13, a chemoattractant important for CXCR5-dependent B cell homing and migration toward the primary follicles (Figure 1) [4,5]. MRCs further express the receptor activator of nuclear factor-κB ligand (RANKL, also known as TNFSF11) [2,5]. RANKL plays an important role in LN organogenesis, and its deletion leads to an absence of LN development [6,7,8]. A recent study revealed that RANKL expression by MRCs activates RANK on LECs and consequently promotes sinusoidal macrophage differentiation [9]. Interestingly, sinusoidal macrophages further cooperate with DCs to activate memory CD8^+^ T cells during viral infection in order to promote antiviral T cell immunity [10]. Moreover, MRCs have been recently shown to be the precursors of follicular dendritic cells (FDCs, CD31^−^gp38^+^, CD21/35^+^), which play important roles in B cell responses in the light zone of GCs (Figure 1) [11]. As MRCs, FDCs produce CXCL13, resulting in the attraction of both B cells and follicular helper T cells (Tfh) toward the primary follicles, where they actively participate in the GC reaction, leading to potent B cell responses and antibody production [12]. B cell, T cell and T–B border reticular cells (corresponding to BRCs, TRCs and TBRCs) are localized in the B cell follicular zone, the T cell zone (paracortex) and the border between the two zones (Figure 1), respectively [13]. Briefly, BRCs and TBRCs express CXCL12 and the survival factor APRIL, supporting plasma cell exit from GCs [13,14]. In addition, together with MRCs, BRCs and TBRCs produce the B cell activating factor (BAFF) known to support B cell maturation and plasma cell survival [2,15]. Focusing on T cell immunity, TRCs are the main populations of FRCs in SLOs and play a major role in naïve T cell homeostasis and survival by producing, together with LECs, the cytokine IL-7 [16,17]. All FRC subsets produce CCL19 and CCL21, which are important for immune cell homing into the LNs [18]. However, among FRCs, two different subsets have been distinguished: one expressing high levels of CCL19 and localized in the T cell zone and another expressing low levels of CCL19 and located at the interface of the T–B cell zone [2]. The production of these two chemokines by TRCs supports T cell localization into the LN cortex as well as their interaction with DCs. Finally, the medullary reticular cells (MedRCs) are adjacent to lymphatic vessels (LVs) in the medullary parts of LNs, where immune cells exit toward the periphery (Figure 1) [15]. 

LN LECs derive from specialized angioblasts [19] and, due to their draining functions in LVs, play a key role in immune cell trafficking (entry and exit from LNs). Similar to FRCs, LECs release CCL21 to promote lymphocyte homing into LNs. Moreover, single-cell mRNA sequencing of LN cells identified four LN LEC subsets in mouse samples, while up to six subsets were defined in human LNs [20,21,22]. LN LEC subsets are so far less characterized compared to FRC subsets. Two LN LEC subsets—the floor LECs (fLECs) and ceiling LECs (cLECs)—localized in the subcapsular parts of LNs and in close proximity to afferent LVs, seem implicated in immune cell entry into LNs (Figure 1) [21]. Another subset is localized in the medullar part of LNs (MedLECs), where immune cells exit from LNs (Figure 1). Finally, a fourth subset localized in the paracortical sinus, adjacent to the medulla, expresses high levels of Spns2 (spinster homolog 2), the transporter required to release sphingosine-1-phosphate (S1P) in lymph (Figure 1) [21]. LEC-derived S1P has been previously shown to play a crucial role in lymphocyte egress from LNs [23]. Together with its paracortical sinus localization, it is tempting to speculate that the LN LEC subset expressing high levels of Spns2 generates a gradient of S1P and therefore regulates (T) lymphocyte egress from LNs. LEC-derived S1P does not only control lymphocyte egress from LNs. A study further revealed that Spns2 expression, and therefore S1P release by LECs, inhibits naïve T cell apoptosis by promoting T cell mitochondrial functions, hence playing an important role in naïve T cell homeostasis and survival [24]. Moreover, the type II receptor of S1P (S1PR2) is highly express by Tfh cells, and genetical deletion of S1PR2 impairs their localization in GCs [25]. However, as S1P is also important for T cell survival, it is still unclear whether this observation is a consequence of a direct impact on Tfh retention in GCs or an alteration of Tfh viability. 

Several LN BEC subsets have also been defined, including the high endothelial venules (HEVs) localized in the cortical part, or in T cell zone, and the postcapillary venules located in the LN medulla, implicated in immune cell entry and egress from LNs, respectively (Figure 1) [26,27,28,29]. By expressing ICAM-1 and CCL21, HEVs allow lymphocyte, especially T cell, extravasation and homing into the LNs [29], therefore playing a major role in immune surveillance. 

Altogether, LNSC subsets play important functions in LN mapping and immune cell organization. This is essential for immune cell homeostasis and functions, such as migration, immune surveillance and cell–cell interactions leading to immune cell activation. Furthermore, LSNCs can also directly modulate immune response. Over the past decade, several studies have highlighted a direct impact of LNSCs on immune responses, especially T cell responses, in different immunological contexts, such as autoimmune disease, viral infections or cancers. 

## 3. LNSCs Function as a Brake for the Development of T Cell-Mediated Autoimmune Diseases

Central tolerance is an important mechanism occurring during T cell development in the thymus and consisting of the deletion of thymocytes expressing a TCR recognizing tissue-restricted antigens (TRAs) with high affinity [30]. Medullary thymic epithelial cells, migratory and thymic-resident DCs present TRAs to developing T cells and therefore play a major role in this process [30,31]. Unfortunately, some autoreactive T cells exit from thymus to populate the SLOs, including LNs, spleen and Peyer’s patches. Additional mechanisms implemented in SLOs inhibit autoreactive T cells in order to avoid their potential activation, therefore maintaining peripheral T cell tolerance. In the last few years, studies have highlighted several ways by which LNSCs contribute to this process. Under inflammatory condition, LECs and FRCs release soluble factors, such as nitrite oxide (NO) and indoleamine 2,3-dioxygenase (IDO), to dampen T cell activation and proliferation (Figure 2A) [32,33,34,35]. Moreover, LECs regulate T cell immunity by altering DC maturation and migration [36,37]. Interactions between DCs and LECs, under inflammatory conditions, lead to the downregulation of costimulatory molecules expressed by DCs, therefore impairing their ability to activate naïve T cells [37,38]. LNSC also contribute to the tolerogenic properties of mesenteric LNs (mLNs). Together with mLN DCs, mLN FRCs express elevated levels of the enzyme responsible for the production of retinoic acid [39], promoting tolerogenic DCs and therefore a Treg-permissive environment. Further studies have shown that FRCs in mLNs undergo a tolerogenic imprinting, which takes place in the neonatal phase [40] and is dependent on the intestinal microenvironment [41].

Moreover, similarly to medullary thymic epithelial cells, LECs, BECs and FRCs have been shown to endogenously express some patterns of TRAs. In mice, the presentation of these TRAs though MHCI molecules to autoreactive CD8^+^ T cells leads to an initial step of proliferation, followed by their deletion (Figure 2A) [42,43,44,45,46,47,48]. A recent study showed that, although most CD8^+^ T cells primed by steady-state LECs undergo apoptosis, a fraction that persists in LNs exhibit transcriptional, metabolic and phenotypic features of long-lasting CD8^+^ central memory T (T_CM_) cells, which preferentially remain in LNs [49]. LEC-educated CD8^+^ T_CM_ cells rapidly proliferate and differentiate into effector cells upon antigenic rechallenge and protect mice against a subsequent bacterial infection [49]. However, this study also indicates that LEC-educated CD8^+^ T_CM_ cells display more stringent reactivation requirements compared to conventional DC-educated CD8^+^ T_CM_ cells and differentiate into functional effectors only when reactivation takes place under Th1-polarizing conditions (LPS + IFNα + IL-12). Altogether, this shows that, while their undesirable reactivation cannot be triggered under homeostatic conditions, LEC-educated T cells contribute to the pool of T_CM_ cells and differentiate into protective effectors upon infection.

LECs, BECs and FRCs express basal levels of endogenously MHCII molecules under the control of the promoter pIV of the MHCII transactivator CIITA, which can be further upregulated in the presence of IFNγ [50]. Murine LNSCs can additionally acquire from DCs peptide–MHCII complexes that they present to CD4^+^ T cell in order to promote their dysfunction (Figure 2A) [51]. MHCII expression by LECs and FRCs has been shown to play important roles in Treg homeostasis in mice (Figure 2A) [52,53]. Our laboratory recently demonstrated that abolition of genetic MHCII expression in murine LNSCs, especially in LECs, impacts Treg proliferation in LNs and leads to the development of spontaneous features of autoimmunity in elderly mice [54].

Due to their tolerogenic role in T cells, LNSCs are important modulators of autoimmune diseases. A deeper understanding of their implications in various autoimmune diseases, such as rheumatoid arthritis (RA), type1 diabetes (T1D) or lupus, is crucial to better elucidate the mechanisms accounting for disease development and to discover novel therapeutic pathways. 

### 3.1. Rheumatoid Arthritis (RA)

Rheumatoid arthritis is a Th1- and Th17-mediated autoimmune disease characterized by joint inflammation, associated with fibroblast-like synoviocyte (FLS) hyperproliferation (pannus formation) and bone/cartilage degradation. Th17 cells play a major role by activating not only macrophages and osteoclasts but also local SCs, such as FLSs and endothelial cells, via their production of IL-17 [55]. When activated in vitro, FLSs produce some chemokines, such as CXCL9/10/11 and CCL2/20, to recruit human Th1 and Th17 cells and express human leukocyte antigen – DR isotype (HLA-DR) molecules to present, at least in vitro, antigens to CD4^+^ T cells [56]. Therefore, an important cross-talk seems to occur between SCs, Th1 and Th17 cells during RA development, leading to the local activation of SCs in joints, and may directly impact autoreactive T cell recruitment and activation.

Macrophages localized in inflamed joints produce high levels of the lymphangiogenic factor vascular endothelial growth factor C (VEGF-C). VEGF-C promotes the proliferation of LECs and the development of LVs in the synovial membrane, increasing the drainage of antigens and autoreactive immune cells from the joints to the draining LNs (dLNs) [57,58]. During the initiation of the disease, the early phase, called “expansion phase”, is characterized by an increased cellularity in popliteal LNs, both in mouse models and in RA patients [59,60,61]. However, experiments performed in mice have further revealed that, during the chronical phase of the disease, LVs may collapse, leading to a local retention of immune cells in joints and an increased disease severity [59,60]. In TNFα-overexpressing transgenic mice, which spontaneously develop arthritis, synovial LECs express increased levels of inducible nitric oxide synthase (iNOS) and produce more NO, inducing an inhibition of LV contraction and consequently a decreased lymphatic drainage from joints to dLNs [62]. Thus, lymphatic clearance plays an important role in reducing the local inflammation in joints; therefore, LV density or VEGF-C levels could be used as biomarkers of disease onset and targeted to dampen disease severity. 

As mentioned before, LNSCs exhibit tolerogenic functions by regulating the activation of autoreactive T cell in LNs. Recent studies have revealed that cultured LNSCs (FRCs and double negative) from RA patients express lower levels of the T cell guideline chemokine CCL19 compared to healthy donors, possibly contributing to aberrant T cell positioning and responses [63]. Moreover, LNSCs from RA patients inefficiently restrain T cell proliferation [64] and express lower levels of HLA-DR compared to healthy donors [65]. Altogether, these data suggest that alterations of LNSC phenotype and functions in RA patients might contribute to enhanced pathogenic autoreactive T cell activation and proliferation. However, the mechanisms responsible for the impairment of LNSC’s ability to regulate T cell responses so far remain unelucidated.

### 3.2. Type 1 Diabetes (T1D)

In Type 1 diabetes, the breakdown of T cell tolerance initiates pancreatic islet β-cell recognition and attack by autoreactive T cells, leading to an impaired production of insulin [66]. Recent studies, performed with both T1D patient samples and T1D-prone NOD (non-obese diabetic) mice, revealed that SCs in pancreatic LNs upregulate MHCII molecules and the coinhibitory marker PD-L1 [67]. Therefore, pancreatic LNSCs exhibit an enhanced tolerogenic phenotype, suggesting that they might function as a brake in the breakdown of islet-specific T cell tolerance. Moreover, NOD mice overexpressing CCL21 in pancreatic β-cells (Lns2-CCL21) display an accelerated formation of local tertiary lymphoid organs in pancreatic islets compared to wild type NOD mice. The FRCs in tertiary lymphoid organs endogenously express higher quantities of pancreatic β-cell antigens in Lns2-CCL21 NOD mice compared to NOD controls [68]. Although a direct antigen presentation by FRCs has not been demonstrated in this context, the overexpression of CCL21 by pancreatic β-cells, together with the increased expression of autoantigens by FRCs, might lead to the induction of local T cell tolerance and a decrease in T1D development. Thus, FRCs and/or CCL21 production by pancreatic β-cells may represent new, interesting therapeutic targets to dampen autoimmune T cell activation in T1D. Finally, a study has shown that IDO-expressing dermal fibroblasts injected intraperitoneally in NOD mice after disease onset migrate to LNs and lead to an increase of Treg concomitant to decreased diabetogenic T cell responses and disease amelioration [69]. Although the mechanisms accounting for T cell tolerance were not elucidated, this suggests that IDO^+^ fibroblasts in LNs might reinstate self-tolerance in NOD mice.

### 3.3. Systemic Lupus Erythematosus (SLE)

Systemic lupus erythematosus is a systemic autoimmune disease affecting several organs, such as the skin, the central nervous system, the joints and/or the kidney, resulting from the loss of T cell and B cell tolerance, immune cell infiltration in peripheral tissues and ultimately organ damage [70]. A recent study revealed that two main signals support the differentiation of plasma cells in LNs and enhance the production of antibodies. The first signal is the production of IL-21 by Tfh cells, and the second one is the production of APRIL, IL-6, CCL19, CCL21 and CXCL12 by FRCs localized at the T–B border [71]. In SLE, it has been further demonstrated that FDCs represent a critical source of IFNα, supporting GC formation and promoting autoreactive B cell activation [72]. IFNα, also produced by plasmacytoid DCs [73,74], has been shown to contribute to the programming of Tfh cells, which are crucial for GC formation [75]. Blocking IFNα using depleting antibody restores B cell tolerance, decreases GC formation and reduces autoantibody production. Thus, by producing IFNα, FDCs support Tfh differentiation and promote autoreactive B cells activation into GC, therefore contributing to the development of SLE (Figure 2A).

## 4. LNSCs Modulate Alloreactive T Cell Activation Following Transplantation

Recent studies have revealed important roles of FRCs in the modulation of alloreactive T cells after cardiac transplantation in mice [76,77]. By producing the extracellular matrix (ECM) inflammatory component Laminin α5 (Lama5), FRCs surrounding the HEVs in the paracortical zone of LNs promote alloreactive T cell activation, leading to graft rejection (Figure 2A) [76]. Genetic abolition of Lama5 expression in SCs results in increased Treg numbers in LNs, providing a tolerogenic environment associated with decreased alloreactive T cell activation and enhanced cardiac allograft acceptance [76].

Moreover, the proinflammatory environment in LNs draining the allograft site induces a FRC senescence-associated secretory phenotype (SASP), characterized by a higher expression of the β-galactosidase (a marker of senescence) and the secretion of proinflammatory cytokines [77]. SASP FRCs overproduce ECM, especially type 1 collagen, leading to LN fibrosis [77]. However, the intravenous injection of healthy FRCs decreases ECM accumulation in LNs and consequently prevents LN fibrosis [77].

After allogenic bone marrow (BM) cell transplantation, alloreactive T cells promote inflammation and may induce a graft versus host disease (GVHD). Acute GVHD, driven by alloreactive T cell activation and associated with an inflammatory process occurring in the skin, gut and lymphoid organs, can rapidly resolve with glucocorticoid treatment [78]. However, during chronic GVHD, the inflammation occurring in several organs may further lead to the development of autoimmune diseases, such as scleroderma or immune cytopenia [79]. A recent study revealed the importance of LNSCs, in particular FRCs, in acute GVHD development after allogeneic BM transplantation [80]. FRCs express DLL1 and DLL4 notch ligands, allowing their cell–cell interactions with allogenic T cells expressing the receptors NOTCH1 and 2 [80]. While the genetic deletion of DLL1 or DLL4 in FRCs has no impact on T cell homing, survival and proliferation, it prevents GHVD development by reducing the number of T cells producing proinflammatory cytokines (such as IFNγ and TNFα) [80].

Another study indicated that allogenic CD8^+^ T cells activated during GHVD induce FRC apoptosis, especially of the TRC population, in a Fas/FasL-dependent pathway [81]. The disruption of TRC networks subsequently damages T cell organization and T cell-dependent humoral responses [81]. Moreover, during GVHD, LN FRCs downregulate DEAF-1, an autoimmune regulator-like transcription factor required for the expression of several TRAs, leading to a breakdown of peripheral tolerance mechanisms dependent on TRA expression by SCs and promoting autoimmune disease development in BM-transplanted mice [82].

Altogether, these studies identified LNSCs as critical modulators of T cell responses during allo- and autoimmunity. The dual role of LNSCs, especially for LEC, BEC and FRC subsets, in impacting peripheral T cell responses appears to be dependent on the immunological context. On the one hand, LECs, BECs and FRCs exhibit tolerogenic functions and dampen autoimmune disease development, while on the other hand, FRCs exposed to an inflammatory environment after allograft promote alloreactive T cell activation and contribute to graft rejection. The mechanisms implicated in these different contexts still need to be better understood to be potentially targeted in new therapeutic pathways.

## 5. Role of LNSCs in T Cell Responses during Viral Infection

CD8^+^ T cells are the main cellular actors controlling peripheral viral infections [83]. Th1 CD4^+^ T cells also play a protective role through the secretion of IL-2 and IFNγ, which further promote CD8^+^ T cell antiviral functions. In contrast, Treg cells generally exhibit detrimental roles during infections, preventing the clearance of viruses by directly suppressing CD8^+^ T cells [84]. The role of Th17 cells in viral immunity is more controversial. On the one hand, the production of IL-17 by Th17 cells inhibit the differentiation of Th1 cells, consequently altering effector CD8^+^ T cell responses [84]. On the other hand, and as reviewed by Ma and colleagues, IL-17 could also limit viral infection-induced organ pathologies [85].

Several groups have reported that virus-associated antigens persist in SLOs for several weeks after the clearance of various viruses (influenza, VSV, vaccinia) in mice [86,87,88]. In particular, Tamburini et al. showed that LECs were capable of acquiring and maintaining viral antigen in LNs for more than 30 days after either vaccinia virus challenge or subunit vaccination [87]. Interestingly, the proliferation of LECs induced by the inflammation was a prerequisite of antigen capture and intracellular storage by LECs. Moreover, antigen archiving in LECs resulted in a greater expansion of adoptively transferred antigen-specific CD8^+^ T cells and a better recall of memory CD8^+^ T cell responses (Figure 2B) [87]. Altogether, these findings highlight LECs as antigen reservoirs capable of promoting protective cellular immunity against viruses. Finally, although LECs that archive viral antigens upregulate MHCII molecules upon infection, it seems that they are not able to directly present these antigens through MHCII but instead transfer them to DCs for presentation to CD4^+^ T cells (Figure 2B) [87,89]. It is, however, possible that other LNSC subsets, such as FRCs, might function as MHCII-restricted antigen presenting cells in the context of viral infection. Indeed, the genetic ablation of MHCII-mediated antigen presentation in radioresistant SCs resulted in an impaired contraction of Ag-specific CD4^+^ T cells, suggesting that antigen presentation by SCs negatively modulates the later phase of the viral specific CD4^+^ T cell responses [90].

An additional role for LECs in controlling cutaneous viral infections has been recently described. Following vaccinia application to skin by scarification, LVs exposed to type I IFN contribute to antiviral T cell responses in LNs and subsequent viral clearance by decoupling fluid and cellular transport, leading to viral sequestration in the skin while maintaining DC transport to dLNs [91].

Several transcriptional pathways, such as the ones implicated in cell division, antigen presentation or chemokine production, are altered in LNSCs following viral infection [89,92]. Among others, the IFN-inducible chemokine CXCL9 is upregulated by the three subsets: BECs, LECs and FRCs. Although the exact contribution of CXCL9 in the eradication of viruses remains to be determined, the CXCL9/CXCR3 axis plays essential roles in the spatial distribution, the migratory behavior and the effector functions of T cells [93,94]. FRCs express a broad range of different molecules and, therefore, promote contradictory effects during viral infection. On the one hand, the upregulation of immunostimulatory molecules, such as ICOSL (inducible T cell costimulatory ligand), CD40 and IL-6, will support the differentiation, effector function and survival of activated CD8^+^ T cells (Figure 2B) [92]. In addition, FRCs localized in LNs reactive to an infection with lymphocytic choriomeningitis virus (LCMV) infection represent a significant source of IL-33, which promotes the expansion and differentiation of CD8^+^ T cells (Figure 2B) [95]. In agreement, the genetic deletion of IL-33 in FRCs results in impaired LCMV-specific CD8^+^ T cell responses both in terms of frequencies and IFNγ and TNFα production [95]. On the other hand, expression of the cyclooxygenase-2 (COX2) and its product prostaglandin E_2_ (PGE_2_) by FRCs upon viral infection result in an inhibition of the T cell response (Figure 2B) [96]. As the modulation of CD28 or IL-2 signaling does not overcome the attenuation of virus-specific T cell responses by FRC-derived COX2 and PGE2, it has been proposed that the pathway directly affects TCR signaling, leading to T cell exhaustion and increasing viral load [96].

Finally, it has been suggested that FRCs could directly trigger lymphocyte infection. First, primary human LN FRC cell lines (hFRCs) can capture HIV-1 (human immunodeficiency virus-1) particles released by infected CD4^+^ T cells, without being themselves productively infected (Figure 2B) [97]. Second, hFRCs enhance the infection of healthy CD4^+^ T cells when cultured with a mixture of HIV-infected and noninfected CD4^+^ T cells (Figure 2B) [97]. The effect is abrogated by the addition of a transwell membrane to the culture, indicating that FRC–T cell contacts are necessary for viral transmission [97]. Therefore, upon encounter, FRCs can transinfect uninfected T cells with viral particles captured from infected T cells, suggesting that FRCs could contribute to HIV-1 spreading in vivo [97].

Therefore, studies elucidating the cross-talk between LNSCs and lymphocytes during viral infection are emerging and will enable a better understanding of how LNSCs modulate viral immunity.

## 6. Cancer-Associated T Cell Responses Are Modulated by (LN-)SCs

In cancer, T cell responses occur to eliminate tumoral cells in order to inhibit/dampen tumor growth. However, tumoral cells have developed several mechanisms to escape from the immune system. For instance, they can decrease their antigenicity and prevent antitumor T cell immune responses by upregulating PD-L1 expression and promoting Treg infiltration. These mechanisms favor a tolerogenic tumoral environment that triggers an inefficient T cell antitumoral response [98,99,100]. Therefore, it is crucial to design therapies that boost antitumoral T cell efficacy to eradicate cancer. Immune check point blockade therapy, such as anti-PD-L1, anti-PD-1 or anti-CTLA-4 (cytotoxic T-lymphocyte antigen-4) antibodies, is currently used to improve T cell activity [101,102].

Mechanical and molecular features of tumor-associated LECs, BECs and fibroblasts modifying host antitumor T cell immunity have been extensively reviewed [103,104,105,106] and will not be detailed in this review. Briefly, cancer-associated fibroblasts (CAFs) are one of the most abundant stromal cell subsets in human cancer. They express several immunosuppressive factors that modulate the tumor microenvironment (TME) and alter antitumor immune responses. They can indirectly inhibit CD8^+^ T cell activation by dampening DC maturation and recruiting Treg cells [105]. Their secretion of soluble factors can also directly dampen CD8^+^ T cell cytotoxic functions or promote their apoptosis [105]. Recently, Lakins and colleagues have shown that CAFs cross-present tumoral antigens to CD8^+^ T cells and induce their death through PD-L2 and FasL pathways [107]. In human breast cancer, CAFs are subdivided into four populations (S1–4) with different repartition, spatial distribution and immunoregulatory functions. Tumors from patient with triple negative breast cancer enriched in CAF-S1 subset display higher Treg cell and lower CD8^+^ T cell proportions compared to CAF-S4-enriched tumors. In contrast to CAF-S4, CAF-S1 attract, retain and favor the differentiation and survival of Treg cells into tumors [108].

In tumor, the blood vasculature is structurally and functionally abnormal, leading to diminished blood flow and restrained oxygen supply, a process called hypoxia. Hypoxia dampens antitumor cytotoxic T cell responses directly by decreasing effector T cell proliferation and differentiation or indirectly through the recruitment of suppressive immune cells, the induction of PD-L1 on myeloid-derived suppressor cells, DCs, macrophages and tumor cells, and the inhibition of DC maturation and migration [104]. The TME further supports effector T cell exclusion from the tumor by hindering the expression of adhesion molecules, such as ICAM-1 and VCAM-1, and inducing the expression of immunosuppressive molecules, such as FasL and PD-L1, at the surface of tumor-associated BECs [103,104]. In human breast cancer, angiogenesis-related genes are associated with good prognosis, reflect vessel normalization and correlate with increased TCR signaling [109]. This observation is extended to other cancer types, in which CD4^+^ T cell infiltration positively correlates with genes associated with good prognosis. Interestingly, in a mouse model of mammary tumor, vascular normalization has been shown to increase the infiltration and function of intratumoral CD4^+^ T cells [109]. In turn, effector Th1 CD4^+^ T cells strengthen the normalization of tumor blood vessels by increasing the coverage of blood vessels with pericytes. Although IFNγ production has been shown to inhibit angiogenesis [110], these results highlight a mutual positive regulation between CD4^+^ T cells and tumor vessel normalization [109].

LVs exhibit both pro- and antitumor effects. LV remodeling induced by soluble tumoral factors contributes actively to the dissemination of metastasis to the dLNs and distant organs [111,112,113]. Broggi et al. observed an enrichment of tumor-derived factors in the lymphatic exudate of melanoma patients with a distinct protein pattern between early versus advanced metastatic stage [114]. Based on these results, the use of the lymphatic exudate composition as biomarker should be considered in the future to help the design of personalized therapies. The lymphatic vasculature is also crucial for the initiation of tumor-specific T cell responses by facilitating antigen transport and DC migration from the tumor to the dLNs [115,116]. However, tumor-associated LVs restrain effector CD8^+^ T cell intratumoral accumulation through the upregulation of PD-L1 molecules [117] and the promotion of Treg suppressive functions locally in the tumor (Gkountidi et al., submitted). Interestingly, VEGF-C-induced lymphangiogenesis potentiates the response to immunotherapy [118,119] by favoring a CCL21-dependent naïve T cell infiltration into the tumor [119]. Altogether, these findings suggest that the dual role of LVs on antitumor immunity might be temporally regulated. Therefore, a time-adjusted targeting of LVs could be an interesting therapy to limit primary tumor expansion.

Although the implication of SCs in LNs has not been extensively investigated in cancer immunology, few studies have described their immunosuppressive functions on antitumor T cell responses. In lymphangiogenic mouse melanoma expressing the model antigen ovalbumin (OVA), LECs from the tumor dLNs induce OVA-specific CD8^+^ T cell dysfunction and apoptosis through MHCI-mediated OVA cross-presentation (Figure 2C) [120]. In the same cancer model, while the genetic abrogation of MHCII in LECs leads to a local impairment of Treg suppressive functions in tumors, it does not affect antitumor CD4^+^ T cell responses in LNs draining the tumor site (TdLNs), suggesting that LECs in LNs cannot present tumor antigens through MHCII molecules (Gkountidi et al., submitted). Another study performed in vitro showed that the coculture of human LN LECs with gastric cancer cells enhanced the expression levels of IDO and PD-L1 in LECs, whereas it dampened their expression of IFNγ-induced MHC-II molecules (Figure 2C) [121]. The authors further showed that CD4^+^ T cells failed to be activated in vitro when cocultured with IFNγ-stimulated LECs [121]. With inflammatory cytokines, including IFNγ, being produced in tumors and tumor-draining LNs, this study suggests that human LN LECs dampen tumor-specific CD4^+^ T cell activation in vivo. However, future studies will decipher how LN LECs affect antitumor T cell responses in tumor dLNs and precisely assess their implication in vivo in cancer immunity. 

Among LNSCs, FRCs also impact antitumor T cell responses. Analysis of FRCs from LNs draining tumor site (TdLNs) and nondraining LNs (ndLNs) in melanoma bearing mice shows that FRCs from TdLNs proliferate significantly more than FRCs from ndLNs, leading to a higher proportion of FRCs in the whole TdLN [122]. However, although more FRCs are observed in TdLNs, the distance between adjacent FRCs is increased, leading to a conduit enlargement associated with a disorganization of the T cell area [122]. In addition, the transcriptional analysis of FRCs from TdLNs compared to ndLNs shows their decreased expression of CCL21 and IL-7 (Figure 2C). The decreased production of these two factors, which are important for T cell homeostasis, trafficking and location into the LNs, correlates with diminished T cell area and decreased CD4^+^ T cell frequency in TdLNs compared to ndLNs (Figure 2C) [122]. Moreover, a loss of margin between the T cell and the B cell zones was observed in TdLNs, suggesting that tumor-driven FRC perturbations alter immune cell organization in lymphoid organs [122]. Decreased IL-7 production by FRCs, as well as impaired FRC density, in TdLNs was confirmed in another cancer mouse model (Lewis lung carcinoma cells) and in LN biopsies from patients with colon cancer and further correlated with impaired CD8^+^ T cell numbers [123].

Altogether, these studies reveal the immunosuppressive functions of LN LECs and LN FRCs on T cell responses in cancer. However, a better understanding of LNSC immunomodulatory functions and implications in T cell responses in cancer will help in the development of strategies targeting LNSCs for the design of immunotherapies. Finally, no study so far has described the implication of BECs localized in LNs (LN BECs) in the modulation of T cell responses in cancer. Given the fact that tumor-associated blood vasculature is a major modulator of the TME and regulates T cell entry and exit from LNs, it will be critical to determine how the TME influences the contribution of TdLN BECs in impacting antitumor T cell immunity. 

## 7. Concluding Remarks

In conclusion, LNSCs are important shapers of peripheral T cell responses. Several studies have highlighted their implication in dampening the activation of autoreactive T cells, promoting the activation of alloreactive T cells. The role of LNSCs during virus infection or in cancer is more controversial, with both the enhancement and inhibition of T cell responses depending on the context.

Most of the studies published so far provide the implication of bulk populations of LECs, BECs or FRCs in the modulation of T cell immunity. However, recent single-cell RNA sequencing analysis has revealed that LNSC populations are highly heterogeneous and can be classified into multiple distinct subsets exhibiting specific phenotypes, LN localization and functions. It is therefore crucial to decipher the contribution of each specific subset to T cell responses in the context of different immune-related pathologies. Future studies will deepen our understanding of the pathways differentially active in individual LNSC subsets and responsible for distinct T cell outcomes. There is no doubt that new therapeutic pathways will be identified to target specific functions of one particular LNSC subset depending on the pathological context in order to either promote or dampen T cell immunity.

## Figures and Tables

**Figure 1 ijms-21-07785-f001:**
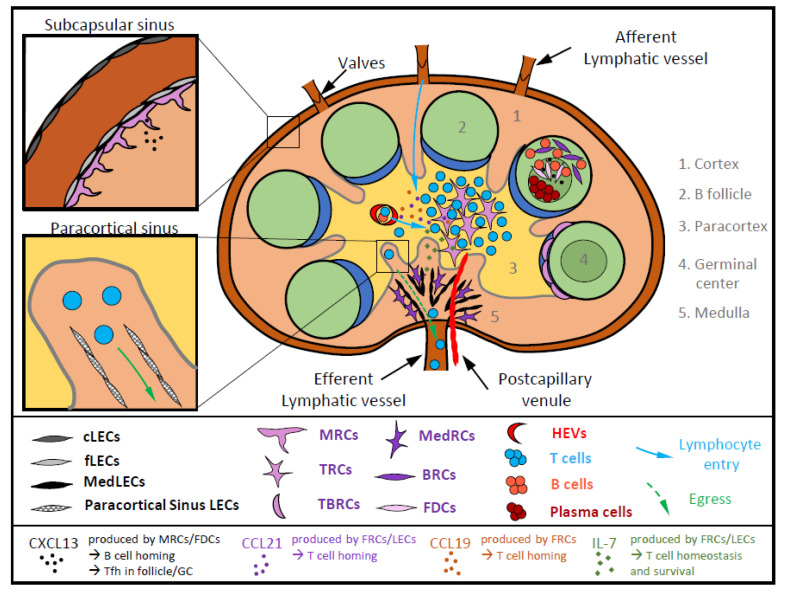
Cartography of different lymph node stromal cell (LNSC) subsets within the LNs. Blood endothelial cells (BECs), fibroblastic reticular cells (FRCs) and lymphatic endothelial cells (LECs) are important components of LNs. Six FRC subsets can be found in different parts of LNs, all producing CCL19 and CCL21, which are important for immune cell homing into the LN. Marginal reticular cells (MRCs) are adjacent to the subcapsular sinus and give rise to follicular dendritic cells (FDCs) in the light zone of germinal centers (GCs). They secrete CXCL13, important for B cell homing and follicular helper T cell attraction in follicle and GCs. B cell reticular cells (BRCs) are also present in GCs, where they are separated by T–B border reticular cells (TBRCs) from the T cell zone (also called paracortex), constituted of T cell reticular cells (TRCs). Medullary reticular cells (MedRCs) are localized in the medullar part, adjacent to efferent lymphatic vessels. LECs, mainly defined as cells lining afferent and efferent lymphatic vessels, have been shown in different locations within LNs and secrete, as FRCS, CCL21. Floor LECs (fLECs) and ceiling LECs (cLECs) compose the outer and the inner part of the subcapsular sinus, respectively. Paracortical sinus LECs are found in paracortical sinuses. Finally, medullary LECs (MedLECs) are in the medullar part of LNs. Both LECS and FRCs secrete IL-7, therefore maintaining T cell homeostasis and survival. BECs form postcapillary venules and high endothelial venules (HEVs), allowing T cells to enter and exit LNs.

**Figure 2 ijms-21-07785-f002:**
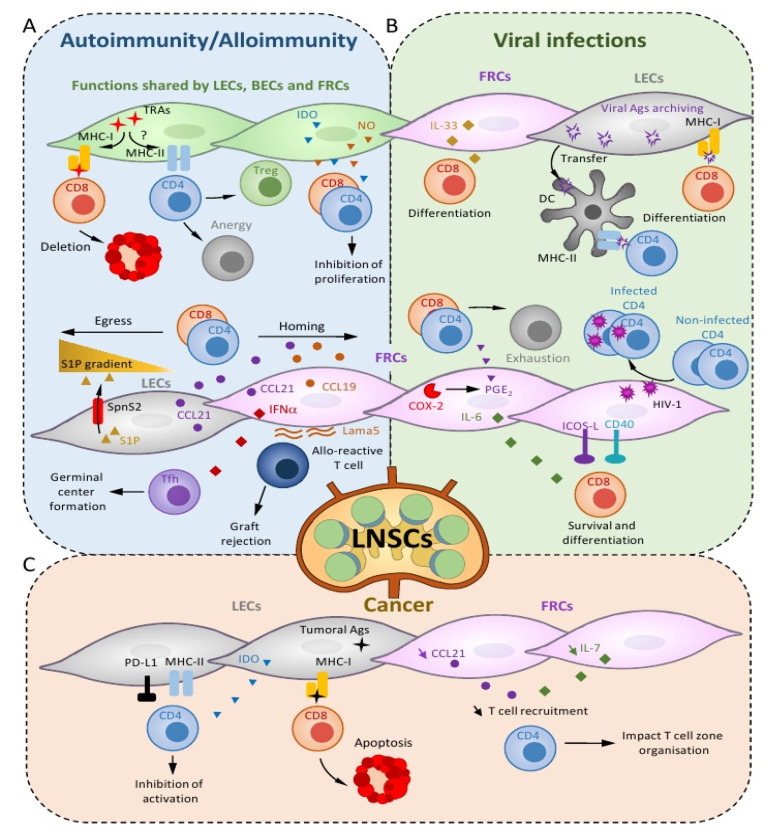
Impact of lymph node stromal cells on T cell immunity. Depending on the inflammatory context, stromal cells in LNs can differentially impact T lymphocytes. (**A**) In autoimmunity/alloimmunity situations, the secretion of soluble factors, indoleamine 2,3-dioxygenase (IDO) and nitrite oxide (NO) by LECs, BECs and FRCs inhibit CD4^+^ and CD8^+^ T cell proliferation. The presentation of tissue-restricted antigens (TRAs) through major histocompatibility complex (MHC) class I (MHCI) by LNSCs to CD8^+^ T cells leads to their deletion. The ability of LNSCs to present TRAs to CD4^+^ T cells through MHC class II (MHCII) is still unclear. Nonetheless, the presentation by LECs, BECs and FRCs of peptide–MHCII complexes acquired from DCs promotes CD4^+^ T cell dysfunction and/or impacts regulatory T cell (Treg) proliferation. The secretion of the chemokines CCL21 and CCL19/21 by LECs and FRCs, respectively, induces T cell homing in LNs. On the contrary, the release of sphingosine-1-phosphate (S1P) by LECs via the transporter spinster homolog 2 (Spns2) generates a gradient that favors the egress of T cells from LN. The production by FRCs of interferon α (IFNα and laminin α5 (Lama5) leads to the formation of germinal centers and the activation of alloreactive T cells supporting graft rejection, respectively. (**B**) During viral infection, viral antigens (Ags) archived by LECs are either directly presented to CD8^+^ T cells or transferred to DCs to be presented to CD4^+^ T cells, inducing their differentiation into effector cells. The secretion of cytokines by FRCs, such as IL-6 and IL-33, or the expression of costimulatory molecules, CD40 and inducible T cell costimulator ligand (ICOSL), promotes the survival and differentiation of CD8^+^ T cells. Moreover, the expression of cyclooxygenase-2 (COX2) and its product prostaglandin E2 (PGE_2_) by FRC following viral infection triggers the exhaustion of T cells. Finally, through the capture of human immunodeficiency virus-1 (HIV-1) particles, FRCs enhance the infection of healthy CD4^+^ T cells. (**C**) In a tumoral context, decreased expression by FRCs of CCL21 and IL-7 correlates with a diminished T cell recruitment, impacting the T cell zone organization within LNs. LN LECs present tumoral Ags through MHCI to CD8^+^ T cells, leading to their apoptosis. They also express MHCII and program death-like 1 (PD-L1) at their surface, leading to CD4^+^ T cell inhibition.

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
