# Peer review of "Lymph Node Stromal Cells: Mapmakers of T Cell Immunity"

_ijms, 2020, doi:10.3390/ijms21207785_

Round 1

Reviewer 1 Report

In this manuscript, Guillaume et al., review and explain the function and regulation of stromal cells (SCs) located in lymph node (LN) in autoimmune diseases, viral infections, and cancer. Moreover, the authors described the architecture of the LN, as well as the LN stromal cells (LNSCs) subtypes and its principal function.

The review cover the most important implications of LNSCs in the biology of the autoimmune disease, viral infections, and cancer, drawing an easy and comprehensive article to follow. The figures improve the quality of the manuscript and assist the reader to follow and understand the complexity of the LN and the LNSCs. Nevertheless, some important molecules or pathways are not described, which could complement and increase the quality and visibility of the review.

Minor suggestions:

  1. Marginal reticular cells (MRCs) in the LN marginal area, are characterized by the expression of the tumor necrosis factor superfamily (TNFSF) member activation of nuclear factor kappa-B ligand (RANKL)(TNFSF11).TNFSF11 is required for LN organogenesis, mice deficient for RANKL or its signaling receptor, lacks all LNs. Considering its implication in the LN this molecule should be described in the review.

  1. The B-cell and T-B border reticular cells, mentioned in the text as BRCs and TBRCs express CXCL12 and APRIL. However another important component, B cell activating factor (BAFF), which support B cell maturation, survival and proliferation, expressed by T cell zone reticular cells is not mentioned. BAFF and APRIL have in common two receptors TNFRSF13B and TNFRSF17 both expressed in B cells and plasma cells. In addition, several studies analyze the potential prognostic effects of BAFF and APRIL in B cell leukemias. Therefore, the authors should try to explain the function and the implication of BAFF expression in the review.

  1. The authors describe the function of S1P and SPNS2 in LN. The antagonist S1PR2 signaling which retains follicular T helper (Tfh) cells in germinal centers is not mentioned. S1PR2 is highly expressed in Tfh cells localize in GCs and genomic deletions of S1PR2 lead to a decrease of Tfh cells in GC. The authors should try to described it because could be interesting as a reader.

  1. Genes involved in MHC Class II antigen presentations are regulated by CIITA. It binds to promoter elements of MHC Class II and IFNγ mediating the their induction. CIITA has also an important role in lymphoma where genomic deletions reduce its expression generating a reduced MHC Class II expression. On the other hand,BLIMP1 a transcription regulator expressed in plasma cells, downregulates CIITA transcription. The authors already address the impact of CIITA in the context of autoinmmune disease but it is also crucial in lymphoma.

Author Response

Reviewer 1:

We thank the reviewer for her/his useful comments that we hope we have adequately addressed. Specifically:

  1. TNFSF expression by MRCs in the LN marginal area and its role in LN development are now described in the revised version.

  1. The role of T cell zone reticular cells derived BAFF has been included in the review.

  1. We are now describing the consequence of blocking S1PR2 on GC formation

  1. “Genes involved in MHC Class II antigen presentations are regulated by CIITA. It binds to promoter elements of MHC Class II and IFNγ mediating their induction. CIITA has also an important role in lymphoma where genomic deletions reduce its expression generating a reduced MHC Class II expression. On the other hand, BLIMP1 a transcription regulator expressed in plasma cells, downregulates CIITA transcription. The authors already address the impact of CIITA in the context of autoimmune disease but it is also crucial in lymphoma”.

Regarding this particular point raised by the reviewer, we have to admit that the literature we have found was related to CIITA genomic alterations and MHCII expression in PMBCL and implication in lymphoma. We could not find any publication related to stromal cells in this context. If the reviewer is convinced that relevant literature should be included, would he be so kind to provide more precision?

Reviewer 2 Report

Very complete extensive review on the role of stromal cells in lymph nodes in T cell immunity. The role is discussed in auto immune disease, viral infection and cancer. The many abbreviations make it difficult to read. Some suggestions:

  1. Maybe add a table with all the different stromal cell types, abbreviation and specificities.
  2. Try and remove abbreviations that are not used many times, or of those of which the use is occasional at larger distance of each other.
  3. Some more details can be added in figure 2, especially the cancer panel: tumoral antigens presented in the context of MHC class I cause apoptosis of CD8 cells, seems strange by itself (or are they specific auto antigens?) but the addition of FasL and PD-L2 expression makes it more logical. The induction of T regs is missing as 

Author Response

We thank the reviewer for her/his useful comments that we hope we have adequately addressed. Specifically:

  1. The reviewer was mentioning the possibility to add a table with all the different stromal cell types, abbreviation and specificities. However, we feel that this would be redundant with the figure 1 legend in which we are already providing this information
  2. We have removed some of the abbreviations
  3. The focus of Figure 2 is to present the contribution of LECs, BECs and FRCs in LNs, and not in peripheral tissues or tumors. This is why the pathways of PD-L2 and FasL haven’t been mentioned. Similarly, the induction of Tregs, and of CD4+ T cells in general, hasn’t been shown for the moment to be impacted by LNSCs (in LNs).